# Religious Filter Bubbles on Digital Public Sphere

**Mónika Andok**

Department of Communication Studies, Pázmány Péter Catholic University, Szentkirályi u. 28,
1088 Budapest, Hungary; andok.monika@btk.ppke.hu

**Abstract:** The aim of the study is to present the online processes related to religious phenomena appearing on digital platforms, primarily the practice of content filtering (gatekeeping, echo chamber, filter bubble), and a critical review of the scientific literature on the field. At the same time, the goal is to create a theoretical introduction to the special issue and a comprehensive examination of the scientific context. For the first time, the study shows that, in terms of media content, filtering can appear from two directions. One is the selections from different events by professional journalists during content creation. The media theoretical literature refers to this aspect as the phenomenon of gatekeeping. Filtering in the other direction takes place on the part of the receivers, who choose from among the available media contents. This phenomenon has already been described by several media scholars, with the concept of selective exposure (Klapper), Daily Me (Negroponte), echo chamber (Sunstein) or filter bubble (Pariser). Focusing on the phenomenon of the filter bubble, the study presents this theory, its criticism and its relevance to religious content and religious communities. The second part of the study focuses on religious filter bubbles and presents the related investigations so far. It analyses in detail the document published by the Catholic Church on 28 May 2023, entitled *Towards Full Presence, Pastoral Reflection on Engagement with Social Media.* During the detailed analytical presentation of the text, the study covers how the opportunities and dangers of network communication and the use of social media appear (including the filter bubble) and what solutions the Catholic Church proposes in this regard.

**Keywords:** filter bubble; filtering; gatekeeping; religious media content; public sphere

## 1. Introduction

The impact of computer mediated communication and social media on social communication, public life and the formation of public opinion has long been a subject of media research. The early analyses of the late 1980s and early 1990s were extremely optimistic, expecting a new, digital agora from Internet communication, on which citizens would rationally discuss their common issues (Arterton 1987; Bowen 1996; Grossman 1995). Towards the end of the 1990s, the optimistic voices died down and the number of critical writings increased, pointing out how Internet communication can have a negative impact on the political and the public sphere (Papacharissi 2002; McChesney 2013; Curran and Seaton 2018; Sunstein 2008; Pariser 2011). In parallel, numerous research groups and individual analyses examined the relationship between religion and the public sphere (Khroul 2014; Rončáková 2017; Tudor and Bratosin 2018; Andok 2018) and the impact of Internet communication on religion (Campbell 2010, 2013; Cheong et al. 2012). After the foundational writings, special attention was paid to the online presence of religious communities (Helland 2000; Campbell and Vitullo 2016; Coman and Coman 2017), the possibilities of online religious identification (Lövheim 2013), and the online transformation of religious life during the coronavirus pandemic (Campbell 2020; Baker et al. 2020). Within this comprehensive scientific horizon, this study deals with the mechanisms and consequences of filtering related to Internet communication, echo chambers and filter bubbles.

In 2023, the Catholic Church celebrated Pentecost on May 28, which according to Catholic teaching is the celebration of the coming of the Holy Spirit. On this day, the Vatican published the document *A Pastoral Reflection on Engagement with Social Media*, in which it draws attention not only to the challenges and possibilities of, the Internet, but also specifically social media. The document clearly refers to the dangers of isolation caused by the filter bubble. "Increasing emphasis on the distribution and trade of knowledge, data, and information has generated a paradox: in a society where information plays such an essential role, it is increasingly difficult to verify sources and the accuracy of the information that circulates digitally. . . . The consequence of this increasingly sophisticated personalization of results is a forced exposure to partial information, which corroborates our own ideas, reinforces our beliefs, and thus leads us into an isolation of "filter bubbles" (Towards Full Presence 2023, sct. 14). The document also indicates that it is not the entire Internet communication itself, but using social media causes the personalized content consumptions that leads to the formation of information and opinion bubbles. In this study, the definition of social media is given and used by Kaplan and Haenlein, while the definition of social network sites (SNS) is based on boyd-Ellison[1]. "Social Media is a group of Internet-based applications that build on the ideological and technological foundations of Web 2.0, and that allow the creation and exchange of User Generated Content. Within this general definition, there are various types of Social Media that need to be further distinguished" (Kaplan and Haenlein 2010, p. 61). Social media is a broader concept, within which we can find social network sites (SNS) in addition to several others (e.g., video sharing sites—YouTube). "We define social network sites as web-based services that allow individuals to (1) construct a public or semi-public profile within a bounded system, (2) articulate a list of other users with whom they share a connection, and (3) view and traverse their list of connections and those made by others within the system. The nature and nomenclature of these connections may vary from site to site" (Boyd and Ellison 2007, p. 210).

In other words, filtering applies especially to those areas of network communication on the Internet that can be linked to the use of web2.0, social media and social network sites. These filtering mechanisms result in polarization, primarily in a political sense, and thus strongly influence the public and democratic functioning. As Sunstein puts it regarding the importance of the public sphere: "The public forum doctrine promotes three important functions. First, it ensures that speakers can have access to a wide array of people. If you want to claim that taxes are too high or that police brutality against African-Americans is common, you can press this argument on many people who might otherwise fail to hear the message. . . . Second, the public forum doctrine allows speakers not only to have access to heterogeneous people but also to the specific people and the specific institutions with which they have a complaint. . . . Third, the public forum doctrine increases the likelihood that people generally will be exposed to a wide variety of people and views" (Sunstein 2008, p. 97).

## 2. Media Content, Media Consumption and Filtering

Filtering related to media content, whether done by the creator of the content or its recipient, is ultimately a way of exercising control. The journalist controls what can reach the receiver, while the receiver selects, filters and thus controls, their own personal reception on social network sites. This type of control is always related to issues of power and authority. That is why American professor Richard Posner rejects filtering in relation to media content creation, equating filtering with censorship. Posner rejects this argument for filtering. First, he says that the argument for filtering is an argument for censorship (Posner 2005, 2 cited by Goldman 2008, p. 115). However, the majority of researchers criticize this view, since this is a massive amount of filtering, but nobody describes such filtering as 'censorship', and such filtering is generally not called an 'infringement' of speech. But then what can we consider filtering in relation to the media? Goldman identifies the potential of the filter in three areas: "Perhaps the standard conception of filtering involves a designated

channel of communication and a system of people with three kinds of roles. First, there are prospective senders, people who would like to send a message. Second, there are prospective receivers, people who might receive messages that are sent. Third, there is a filterer, or gatekeeper, an individual or group with the power to select which of the proffered messages are sent via the designated channel and which are not" (Goldman 2008, pp. 115–16). With regard to religious media content, we can encounter all three types, only the system of criteria for filtering will be different, depending on whether the sender, recipient or gatekeeper is a religious or church person or not.

### 2.1. Filtering Related to the Creation of Media Content

Regarding media contents, the practice of filtering can be grasped from both the side of creation and reception. The conventional news media also employ filtering techniques. Traditional mass communication, more specifically, professional media content producers and journalists, have been filtering content since the middle of the 19th century, when the first news agencies appeared. In other words, they chose what information was published in the columns of each page and what was not. In the selection and filtering, the journalists were guided by the news value and the factors influencing the news worthiness: the unexpectedness of the event, the degree of its drama, the degree of norm violation, the relevance of the event, etc. This phenomenon, journalistic filtering, was named gatekeeping by David Manning White in 1950. The concept of gatekeeping entered the literature dealing with the organizational sociology of journalistic work with White's study (White 1950). It is an apt term, as it vividly captures the selection task that journalists perform when they select news that is considered important or interesting for their readers from the huge amount of news. Here, the journalist guarded the gate to the public and only allowed authentic, verified information through his filter. This selection process has been transformed at several points in digital journalism and, in a broader sense, in the world of social media. On the one hand, content consumers have become content producers themselves, but the majority of the posts they contribute are not public but private, and checking the authenticity of the content is not part of the process of publishing or posting.

With regard to gatekeeping, the literature on media theory predominantly focuses on the selection aspect of the process, but we also find other approaches. Shoemaker summarized these in her 1991 volume, and classified the research up to that time into five large groups. She called the first level of media content filtering the individual level, and included analyzes related to the personality of the gatekeepers, the values they hold and their perception of their role. At the second level, the routines of the gatekeeping process are revealed, as well as what practices and patterns exist in this regard among media employees (e.g., what are the factors influencing news worthiness). Thirdly, at the organizational level, they try to capture what internal formal or informal rules operate in each media organization during the selection process. At the fourth, institutional level, the researchers deal with how the media institution's focus on the external effects of information control affects the gatekeeping process. External influences mean the company's market positions, market value and relationship with its political allies. And finally, Shoemaker mentions the social level as the fifth one, whose analyses reveal the effects of gatekeeping on culture and the dominant political ideology (Shoemaker 1991).

I believe that the aspects of network gatekeeping can also be adapted to religious communities, both on the basis of the subjective factor and the characteristics of the information, external constraints, characteristics of the organization and gatekeeping processes, as well as in terms of the institutional and social environment. We can therefore say that on the social network sites where church organizations and communities share their content, the above-listed filtering criteria on the part of the content creators' work.

In her study, the Israeli researcher Karin Barzilai-Nahon compared the classical theory of gatekeeping with the processes taking place on the network, and thus outlined a meta-theoretical framework for the phenomenon of information control realized during gatekeeping. Barzilai-Nahon describes two types of gatekeepers and mechanisms in the

network, while accounting for the elements and processes that are characteristic of network gatekeeping (Barzilai-Nahon 2005). Barzilai-Nahon also identifies two new roles in terms of network gatekeeping, one is the virtual community provider (VCP) who is present in a kind of care-operating role, while the other is the virtual community manager (VCM). The gatekeeper authority of the Virtual Community Provider extends to all virtual communities using the platform, but at the same time they have no direct relationship with the members of the communities. The authority of the Virtual Community Manager extends to one specific virtual community and has a personal relationship with the members of the community. In the former case, there is no direct flow of information between the virtual community provider and the members, while in the other case there is. (Barzilai-Nahon 2005, p. 18). Regarding the comparison of the two types, we can say that the gatekeepers of religious communities on social network sites are more likely to be classified as Virtual Community Managers.

### 2.2. Reception and Filtering of Media Content

Filtering related to media content can be grasped not only from the side of creation, but also from the side of reception. In other words, newspaper readers, radio listeners, TV viewers and network communication users also select and filter media content. The content filtering described from the receiver's side goes beyond the differences in meaning construction and meaning creation described by Hall (Hall 1980). It is worth mentioning that the functioning of receiver filtering has been identified in each historical stage of mass and network communication. Joseph Klapper developed the theory of selective exposure in 1960, in the era of classical mass communication (printed press, radio, television). In the early period of network communication, before the World Wide Web, Nicholas Negroponte writes about the Daily Me in his book Digital Being. Cass Sunstein's idea of online echo chambers and Eli Pariser's book on the filter bubble was published in 2011. But let us go in chronological order!

In the era of traditional mass communication, in 1960, the theory of selective exposure was described by Joseph Klapper. As an American sociologist, Klapper (1917–1984) searched for the answer to why the effect of media content is limited. Following his studies, Klapper showed that people pay attention or pay more attention to the content of mass communication that fits their existing views, and pay less attention to or outright reject content that is in opposition to their beliefs. In terms of the concept of persuasion, this can be described as the fact that mass communication is generally more likely to confirm people's existing views and less often to change them (Klapper 1960). This theory relies heavily on the ideas of Leon Festinger, who published his theory of cognitive dissonance in 1957. Klapper talks about selection on three levels. On the one hand, the audience does not follow media outlets whose attitudes do not match their own. On the other hand, even if the opposite message reaches the receiver, he or she does not elaborate but simply ignores it. And thirdly, memory selection comes into play, i.e., even though the message with the opposite attitude reached the receiver and he or she accepted it, his or her memory selects it in the short term (Klapper 1960). The functioning of selective exposure was then confirmed by numerous empirical studies (Singleton 1969; Gaddy 1984; Entman 1989).

Nicholas Negroponte, the founder of the MIT Media Lab, already raised the idea of recipient filtering and selection related to network communication in his book *Being Digital*, published in 1995. In his book, he describes one of the Media Lab's projects, the summation of which is that people will not read newspaper content in the classic way on network communication interfaces, but in a personalized way, based on settings according to their own preferences. And this personalized media product was called *Daily Me* by Negroponte. "Imagine a future, in which your interface agent can read every newswire and newspaper and catch every TV and radio broadcast on the planet, and then construct a personalized summary. This kind of newspaper is printed in an edition of one. . . . call it the *Daily Me*" (Negroponte 1995, pp. 153–54). According to Negroponte, the intelligence that compiles the personalized newspaper can be seen in action in two places in the system.

It can be in content creating like having a journalist at home—like tailoring *The New York Times* to your needs. In this case, the bits are filtered, prepared and delivered to us to be printed at home or read in some other interactive way. The other is when the editorial office resides in the receiver, when *The New York Times* transmits a huge amount of bits, up to five thousand different pieces of news, from which our own device selects a few according to our needs, habits and plans for the day. In this case, the intelligence is in the receiver, and the transmitter indiscriminately sends every bit to everyone. The future is neither one nor the other, but both (Negroponte 1995). With *Daily Me*, individuals are customizing and personalizing their news feeds.

After the turn of the millennium, in 2008, when social network sites were just in their infancy, Harvard Law School professor Cass Sunstein introduced the concept of echo chamber characteristic of digital surfaces and digital communities in his book *Republic.com 2.0*. His thought starts from the fact that, in order for members of communities and society to be able to talk about their common issues, it is fundamentally necessary for their experiences to be shared. Without this, without the common ground, it is impossible to carry on a rational discussion. "In a heterogeneous society, it is extremely important for diverse people to have a set of common experiences. Many of our practices reflect a judgment to this effect. National holidays, for example, help constitute a nation, by encouraging citizens to think, all at once, about events of shared importance" (Sunstein 2008, p. 104). A heterogeneous society without shared experiences is much less able to deal with social problems. But network communication and the personalization of news consumption make it possible that we do not have to encounter topics and opinions that we did not choose ourselves. Without any difficulty, we see exactly what we want to see, no more, no less. . . People lock themselves in their own echo chambers, writes Sunstein in the introduction to his book. "Not surprisingly, many people tend to choose like-minded sites and like-minded discussion groups. . . . With a dramatic increase in options, and a greater power to customize, comes an increase in the range of actual choices. Those choices are likely, in many cases, to mean that people will try to find material that makes them feel comfortable, or that is created by and for people like themselves. This is what the 'Daily Me' is all about".—refers back to Negroponte's thinking (Sunstein 2008, p. 99). The American professor is primarily interested in the existence of echo chambers and the resulting polarization for political reasons, and is investigating the phenomenon because of their harmful effect on democratic functioning. "Web sites might use links and hyperlinks to ensure that viewers learn about sites containing opposing views. A liberal magazine's Web site might, for example, provide a link to a conservative magazine's Web site, and the conservative magazine might do the same to a liberal magazine's Web site. The idea would be to decrease the likelihood that people will simply hear echoes of their own voices" (Sunstein 2008, p. 108).

Following Sunstein's theoretical work, several empirical studies confirmed the existence of the echo chamber (Jasny et al. 2015; Baumann et al. 2020).

### 3. The Concept of the Filter Bubble

In 2011, Eli Pariser published his summary volume on the polarizing effect of social media, this effect is caused by personalized media consumption and filtering. "Personalization is based on a bargain. In exchange for the service of filtering, you hand large companies an enormous amount of data about your daily life—much of which you might not trust friends with. These companies are getting better at drawing on this data to make decisions every day (Pariser 2011, p. 10). He defines the filter bubble itself in several ways: filter bubble is a "personal ecosystem of information that's been catered by these algorithms to who they think you are. . . a unique universe of information for each of us" (Pariser 2011, p. 9). Pariser grasps the phenomenon on a broader horizon than Sunstein, because he is not only interested in political divisions. He sees exactly that the filter bubble's costs are both personal and cultural. There are direct consequences for those of us who use personalized filters (and soon enough, most of us will, whether we realize it or not). And there are

societal consequences, which emerge when masses of people begin to live a filter-bubbled life (Pariser 2011, p. 9).

The American author feels that it is extremely problematic that while consumer trust in news agencies is decreasing, the consumption of news created by amateur receivers is increasing. He identifies two problems in this regard: "First, by definition, the average person's Facebook friends will be much more like that person than a general interest news source. This is especially true because our physical communities are becoming more homogeneous as well—and we generally know people who live near us. . . . Second, personalization filters will get better and better at overlaying themselves on individuals' recommendations" (Pariser 2011, p. 23).

> A number of studies and empirical research deal with the decline of reader and receiver trust in news, which show that trust in the institutional system of the media has also decreased. First of all, citing the data of the 2023 Edelman Trust Barometer, we can say that 42 percent of the receivers and readers believe that the media provides misleading information (Edelman Trust Barometer Global Report 2023, p. 10). In addition, the University of Oxford and the Reuters Institute have been publishing comprehensive reports on news consumption trends in 46 countries around the world for more than ten years. According to the Digital News Report (2022), the highest level of trust in news and news sources was measured in Finland at 69%, while the lowest was in the United States and Slovakia at 26%. (Newman et al. 2021, p. 15)

Pariser in his research also shows that the filter bubble not only affects what news people read or do not read, but also determines how they think about the world. Similar to Sunstein, he explains that "living" in different bubbles means that people have a different experience of the world around them, their environment. "This distorting effect is one of the challenges posed by personalized filters. Like a lens, the filter bubble invisibly transforms the world we experience by controlling what we see and don't see. It interferes with the interplay between our mental processes and our external environment" (Pariser 2011, p. 27). Filter bubbles provide new ways of managing perceptions. This then leads to the fact that the filter bubble tends to dramatically amplify confirmation bias, and because the filter bubble hides things invisibly, we are not as compelled to learn about what we do not know (Pariser 2011, p. 29). According to Pariser, customization hinders creativity and innovation in three ways. On the one hand, it narrows the solution horizon, i.e., the pool of ideas from which users can draw to solve common social problems. On the other hand, there will be a lack of key elements in the preparations that would encourage creativity, and finally the filter bubble makes people more passive, since they mostly meet with agreement (Pariser 2011, p. 30). In this regard, the American researcher cites the father of the cultivation theory, George Gerbner, who studied the impact of television in the United States from the 1970s. He found that those who watch television for more than four and a half hours a day will believe in the television reality instead of their external reality, and because of the violent content they see in it, they make the world around them more dangerous. In the case of the filter bubble, it is the other way around, not only is it a mean word, but it creates a particularly friendly world around the user. "Friendly World Syndrome. . . Gerbner called this the mean world syndrome: If you grow up in a home where there is more than, say, three hours of television per day, for all practical purposes, you live in a meaner world—and act accordingly—than your next-door neighbor who lives in the same place but watches less television. If television gives us a "mean world", filter bubbles give us an "emotional world" (Pariser 2011, p. 46).

Pariser also discusses why the period of traditional mass communication was a better media environment in this respect: "Traditional, unpersonalized media often offer the promise of representativeness. A newspaper editor is not doing his or her job properly unless to some degree the paper is representative of the news of the day. This is one of the ways one can convert an unknown unknown into a known unknown. If you leaf through the paper, dipping into some articles and skipping over most of them, you at least know

there are stories, perhaps whole sections, that you passed over. Even if you don't read the article, you notice the headline about a flood in Pakistan—or maybe you're just reminded that, yes, there is a Pakistan. In the filter bubble, things look different. You don't see the things that don't interest you at all" (Pariser 2011, p. 33).

> Producers, content creators of traditional media content, especially news, strove for an authentic, balanced presentation and representation of reality. And in the case of media content edited in this way, the readers and receivers also obtained information about which events were not among their primary preferences and were also confronted with opinions that did not match theirs.

In a later text, Pariser compares the filter bubble metaphor to the membrane, more precisely he writes that the filter bubble is not a fence, but a membrane. And it is easier for advertisers with a commercial purpose to pass through this membrane than for public information. "As the filter bubble's membrane becomes thicker and harder to penetrate, averters could become a powerful adaptive strategy" (Pariser 2011, p. 59).

The filter bubble also plays a role in the fact that, despite social media being expected to function more democratically, it would be less manipulative or open to propaganda than traditional mass communication. The opposite happened, the filter bubble is primarily controlled by a few centralized companies, it is not as difficult to adjust this flow on an individual-by-individual basis as you might think. "Rather than decentralizing power, as its early proponents predicted, in some ways the Internet is concentrating it" (Pariser 2011, p. 43). The filter bubble concept, and the polarization named as a consequence of the phenomenon, found a particularly strong echo among researchers of political communication. Countless empirical studies have confirmed the existence of filter bubbles (Bozdag and Van Den Hoven 2015; Haim et al. 2018; Klug and Strang 2019), on the one hand, and their role in increasing political polarization (Spohr 2017; Barberá 2020).

Now let us think about the consequences of personalized media consumption—which is the basis for the creation of the filter bubble—for religious media consumers.

On social media platforms, user-implemented filtering and personalized filtering come from the determination of private interests, private worlds and privacy. And this (since the Age of Enlightenment) from the point of view of religious beliefs pushed back from public life into private life also created an opportunity. On the one hand, it has enabled many to display their religious beliefs in digital public sphere, thus making religious belief public and publicly observable. On the other hand, religious users specifically search according to their own filtering settings, and are particularly open to religious information. This even means that compared to the pre-digital world, more religious information reaches those who are looking for it, since there is a lot of religious content on the Internet. With regard to other religions, the Pew Internet & American Life Project reports that studies of various religious communities in the US indicate that about 82 million Americans (64% of American users) use the Internet for religious purposes (Hoover et al. 2004).

Network communication and social media have changed not only the religious experience connected to private life, but also the public visibility of church institutions. Religious institutions, as social institutions, were in a marginalized position, but they were also present in (Habermasian) bourgeois public spheres. According to classical media logic, in the mainstream mass communication, more negative than positive news and scandals regarding the churches were included in the media content, often heavily politicized (Schofield Clark and Hoover 1997). In the case of filter bubbles, these negative news can still filter through the membrane of non-religious bubbles, but mainly for the members to see what they disagree with. In some cases, content consumers in non-religious bubbles are also open to more positive or neutral news. Such an event was the papal visit to Hungary in 2023.

*Criticism of the Concept and Theory of Filter Bubbles*

Since the appearance of the filter bubble theory, there have been not only followers but also critics among media researchers (Arguedas et al. 2022). From the last three years, I

highlight two significant ones, one by Dahlgren, the other by Bruns (Dahlgren 2021; Bruns 2021). Critics do not doubt the existence of the basic process itself. They also believe that it is true that during network communication, on the one hand, users look for information that confirms their existing views, attitudes, opinions and beliefs, and they specifically look for supportive content. On the other hand, social network site providers and search engines also work on the basis of algorithms that further strengthen content that supports users' beliefs and does not question them. But in addition to this, Pariser's theory has been criticized on several points. Swedish media researcher Peter Dahlgren lists nine counterarguments in his article regarding the filter bubble (Dahlgren 2021). These are as follows: 1. Filter bubbles can be seen at two levels: technological and societal, 2. People often seek supporting information, but rarely avoid challenging information, 3. A digital choice does not necessarily reveal an individual's true preference, 4. People prefer like-minded individuals, but interact with many others too, 5. Politics is only a small part of people's lives, 6. Different media can fulfil different needs, 7. The United States is an outlier in the world, 8. Democracy does not require regular input from everyone, 9. It is not clear what a filter bubble is (Dahlgren 2021, p. 16).

Dahlgren proposes discussing the functioning of the filter bubble on two separate levels. One is the level of technology, which he considers the level of the individual user, and the other is the social level. He does not deny the phenomenon, identifiability and basic operating mechanism of the filter bubbles themselves, in fact: "These filter bubbles emerge when users actively seek information and the Internet services learn what users consume. The Internet services then try to provide users with more of the same content during their next visit, based on predictions from past behaviours" (Dahlgren 2021, p. 17). And he also sees that all this leads to a spiraling strengthening of self-affirmation in the case of users and can cause narrower self-interest; overconfidence; dramatically increased confirmation bias; decreased curiosity; decreased motivation to learn; fewer surprises; decreased creativity and ability to innovate; decreased ability to explore; decreased diversity of ideas and people; decreased understanding of the world; and a skewed picture of the world (Pariser 2011; 106 cited by Dahlgren 2021, p. 17). The Swedish researcher's counterarguments are primarily theoretical, but in some cases also refer to empirical research. His first thought concerns the fact that filter bubbles can be seen at two levels: technological and societal. In the case of research showing its social impact, Dahlgren also found one that disproved the functioning of the filter bubble (Dahlgren 2021, p. 19). According to his second counterargument, people often seek supporting information, but rarely avoid challenging information. "Selective exposure research has already shown that people, on average, prefer supporting information to challenging information... The evidence for the claim that people avoid challenging information, on the other hand, is much bleaker. This is because people have two motivations: to seek information (which is a moderately strong motivation) and to avoid information (which is a comparatively weak motivation)" (Dahlgren 2021, pp. 19–20). According to his next counterargument, digital choice does not necessarily reveal an individual's true preference. Pariser's idea that people's preferences guide their choice of content: "identity drives media", this argument, according to Dahlgren, follows the interpretation of technological determinism and behaviorism. We believe that Dahlgren is probably right about this, but this is not proof that the idea of technological determinism is wrong. "There are also several theoretical reasons why preferences and choices should be kept separate... We (or the algorithm) can directly observe what a person selects, but we can never directly observe what a person prefers. ... people can make choices that are consistent with their commitments rather than with their preferences (an atheist can visit a religious web site in order to find counterarguments in a debate)" (Dahlgren 2021, p. 21). An important counterargument to Dahlgren is that although people prefer like-minded individuals, but interact with many others too. "... personalization sometimes increases the amount of challenging information... In sum, audience fragmentation is likely to be an accidental, rather than an essential, property of social networking sites" (Dahlgren 2021, p. 22). We also have to agree with the counterargument that politics is

only a small part of people's lives. The filter bubble thesis focuses almost exclusively on political discussions and information. This implies that politics is a large part of people's lives, even though this has been refuted by several empirical studies. (Hill and Dunbar 2003; Eveland and Hively 2009 cited in Dahlgren 2021, p. 24). Active user and politically engaged user are not synonymous, if we consider them as such, we fall into the error of sampling bias. His sixth counterargument is that it is a misconception of Pariser and the filter bubble that people today only get information from social media. Different media can fulfil different needs. Dahlgren uses data from 2022 to prove that people are getting information from other media as well. And this is confirmed by the European Union's 2023 report, The European Media Industry Outlook and the 2022 data of the Digital News Report (The European Media Industry Outlook 2023; Digital News Report 2022).

Dahlgren summarizes his insight like this: "audience fragmentation happens not only within a medium, but between media as well" (Dahlgren 2021, p. 25). In his next counterargument, he emphasizes that the filter bubble's strong political polarizing effect is primarily characteristic of the United States, and that the filter bubble is not a cause but a consequence of polarization. Probably not only in the USA, but where there is a two-party system, and strong political polarization existed even before the advent of the Internet (Dahlgren 2021, p. 26). His eighth counterargument concerns the normative democratic functioning implied by the filter bubble theory. He believes that democracy does not require regular input from everyone. The filter bubble concept is interpreted by Dahlgren as a normative approach, and this has already characterized the theory of selective exposure, but there is no evidence that democracy works only in this way. His last objection concerns the lack of a precise definition of the filter bubble, which the many metaphorical descriptions cannot replace. And he also points to a strong paradox in relation to the idea: "It is an inherent paradox that people have an active agency when they select content, but are passive receivers once they are exposed to the algorithmically curated content recommended to them" (Dahlgren 2021, pp. 28–29).

The other critic of the theory is Bruns, who, in addition to articles, also devoted a separate summary volume to the topic (Bruns 2019, 2021). In these, he admits that the self-reinforcing feedback loop works in social media, but he believes that the filter bubble causes moral panic rather than a real problem (Bruns 2021, p. 33). Like Dahlgren, he highlights the lack of accurate definition and the interchange of cause and effect in the theory. In other words, there was political polarization before, which is rather reflected in the filter bubble, and it is not online filtering that causes political division.

In agreement with the critical approaches, let us now see if we can talk about the functioning of the filter bubble in the case of religious content and religious communities.

In his first critical comment, Dahlgren suggests that the filter bubble effect should be thought of on two levels: one is the level of technology, which exerts its most powerful effect in individual settings, while the other is the social level. When examining the religious filter bubble, we must also propose a third level: the middle level, the level of religious communities, between the personal/technological/micro level and the social/macro level. In other words, an analysis of how the filter bubble effect works in religious and other communities. There are two aspects to consider in this matter. One is that the majority of users belong not only to one group or community on social network sites, but also to several. The other is that the groups and communities themselves must also be differentiated according to types: family, friend, professional, fan, customer groups, etc. The level and intensity of involvement, belonging to the group and whether the commitment is short or long-term must also be taken into account (Harfoush 2016).

The members of religious communities mostly have a long-term and deep commitment, while they can also be members of groups that they join only temporarily. There may be thematic groups (political, fans) that filter strongly, because the members are characterized by a strong, long-term commitment. And there may be groups (customers, problem solvers, e.g., WAZE) that do not filter strongly, because the commitment of the group members is temporary, the group membership means some kind of advantage, in the dimensions of

reciprocity, assistance and cooperation. Overall, we would supplement Dahlgren's first point of criticism in the case of the religious filter bubble by saying that the filter bubble effect should also be examined at the community level, taking into account the types of groups and belonging to multiple groups. Dahlgren's second critical point is also true for religious communities, they actively seek supporting information, but they do not keep away from other content either (Laney 2005; Brubaker and Haigh 2017; Riezu 2014; Ratcliff et al. 2017; Andok 2021, 2023). Dahlgren's third critical point is that online choice does not necessarily reflect personal preferences. Here we have to say that it still reflects personal preferences regarding religious content, on the one hand the search for religious content, and on the other hand the search for content close to the user's religiosity.

The fourth aspect, that users are not only in contact with like-minded people, is also true for religious communities both offline and online. They are not political communities, although there may be overlaps with them—therefore the fifth aspect is also true for religious communities. Dahlgren's sixth point of criticism is that people get information from a variety of media, not only from online interfaces, and this holds true for religious people as well. The other counterarguments that Dahlgren raised against the filter bubble theory have little relevance to our thinking about the religious filter bubble.

All in all, we can say that, in addition to the technological, personal and social levels, a community level should also be included, and religious communities should be viewed as opinion bubbles. Also, we must emphasize that in the case of religious users, online choices regarding religious content reflect real personal preferences.

## 4. Religious Filter Bubbles

Moving forward in the thought process of the study, we will now examine what happens when the filter bubble is not created along political but religious values and commitments. Let us see how this issue, the issue of filters based on religious values, appears both in terms of technology and content. Empirical research on religious media users has shown that religious beliefs will influence media consumption, and a kind of technological and content filtering will also appear (Hoover 2006; Campbell 2010, 2013; Cheong et al. 2012). Religious media consumers choose media content based on the requirements of their church and religion. Of course, this is not universal legality. On the one hand, it depends on the religion to which the media consumers belong, what kind of media consumption they have. On the other hand, it also depends on the receivers themselves, on the extent of their religious commitment, how much they take into account and observe these regulations. In this regard, in the literature on media theory, we find two important interpretive frameworks. One is Barzilai-Nahon and Barzilai's cultured technology framework, the other is Heidi Campbell's Religious Social Shaping of Technology framework (Barzilai-Nahon and Barzilai 2005; Campbell 2010, 2013). Both theories confirm that the question should not be asked in such a way that if culture is part of technology or vice versa, but it is worth starting from the fact that the person using technology definitely belongs to a certain culture or religion.

Karin Barzilai-Nahon and Gad Barzilai look at the Internet as a central phenomenon of contemporary modernity that interacts with practiced fundamentalist religious traditions. In their 2005 paper, Israeli researchers developed the concept of cultured technology, and analyzed the ways communities reshape technology and make it as part of their culture. They presented a theoretical framework to understand the relationship between religious fundamentalist communities and the communication technology, the Internet order to demonstrate how the Internet has been culturally constructed, modified and adapted to the community's needs and how the religious community has been affected by it. They analyzed four dimensions of religious communities: hierarchy, patriarchy, discipline and seclusion. Their specific empirical investigation took place in Jewish fundamentalist communities, and in connection with this they also defined the concept of religious fundamentalism. In popular culture, the term religious fundamentalism often denotes political extremism, violence and terrorism. However, they are dealing with much

broader aspects of religious fundamentalism. On a fundamental level, religious fundamentalism is essentially an ultraconservative approach to religious texts through an attempt to avoid pragmatic compromises with modernity (Barzilai-Nahon and Barzilai 2005, p. 4). According to their definition, religious fundamentalism is a system of absolute values and practiced faith in God that firmly relies on sacred canonical texts. A significant level of affinity among its members, seclusion from the world that surrounds it, strict communal discipline and a patriarchal hierarchy often characterize it (Barzilai-Nahon and Barzilai 2005, p. 2). But it is worth considering that the theory of cultured technology applies not only to fundamentalist communities, but is true for most religious communities. In this case, the Internet by means of complex communal processes that adapt the Internet to fulfil their fundamentalist religious needs and in the process transform it into a new and different type of technology that suits their community. The Barzilais' first point of investigation was hierarchy. Religious fundamentalist communities are characterized by a tight hierarchy (Barzilai-Nahon and Barzilai 2005). The Internet can be culturally constructed in ways that adapt it to the needs of a religious fundamentalist hierarchy. Technology, and specifically the Internet, serve the hierarchical realm well by means of personalization and contextualization tools. For example, the elite may utilize various technologies and information systems to disseminate personalized information to targeted users for purposes of communal socialization and mobilization. The community is able to affix the hierarchical order online, not less than offline, by offering its members virtual services (e.g., e-prayers and online consultations with higher religious authorities) that before the Internet were available only face-to-face (Barzilai-Nahon and Barzilai 2005, p. 7). Their second aspect of analysis was patriarchy, discussing the issue of which they examined the involvement of women in the use of network communication. "In conclusion, we assert that the Internet has created better opportunities for feminine voices to be heard in religious fundamentalist contexts, although these opportunities are framed within the communal context and its hierarchy" (Barzilai-Nahon and Barzilai 2005, p. 10). Third dimension of tension was the discipline. Fundamentalist religious communities are highly disciplined, their behavior being based on sacred texts and the hermeneutics that surround them. Discipline is perceived as the only alternative to blasphemy. A significant change is possible only if the spiritual authorities define a path of hermeneutics that legitimizes it (Barzilai-Nahon and Barzilai 2005, p. 10). The Israeli researchers also approached the issue from the perspective of media history: Contrary to printed texts, Hypertext is interactive, non-linear, associative, not-fixed, modular and not necessarily owned by an identified single author. Many religious fundamentalist communities that are present on the Internet enhance their communal discipline by using applications such as discussion groups, Intranets, list serves, chats and forums that enable collaboration by many users to empower the communal consciousness. The seclusion was the fourth dimension of tension in Barzilais' work. The Pew Internet & American Life Project study (Larsen 2001) reports that most religious surfers (67%) use the Internet to gather information about their own faith and not to learn about other religions. This aspect is particularly important and should be emphasized from the point of view of the filter bubble, because it shows that users are interested in their own religious community, while others are not. They examined the above-mentioned four aspects in the case of ultra-orthodox Jewish communities.

The other major theoretical framework, which is intended to show how religious beliefs and values influence users on social network sites, how they filter, is attributed to the American researcher Heidi Campbell and is called Religious Social Shaping of Technology (RSST) model (Campbell 2006, 2010, 2013). Since the 1960s, many schools of media theory have been dealing with the question of how changes in communication technology affect society, communities or even culture. The most prominent of these is the Toronto School, the theory of technological determinism, which was developed by Marshall McLuhan and is well known in media research. The wide-ranging and influential theory actually generated critical positions right from its appearance. One of these is the theory of social-shaping of technology (SST). The social-shaping of technology research also assumes that

the dominant (communication) technology of a given era is decisive in relation to society and culture, but with a different explanation than technological determinism (Williams and Edge 1996). While the latter explained the change from the point of view of technology, SST, speaking from the point of view of the community adapting the technology, states that the community will determine the direction and extent of technological innovation (MacKenzie and Wajcman 1999). This theory was further developed by the American Heidi Campbell, specifically for the use of media technology by religious communities (Campbell 2006, 2010). Campbell researches how religious groups, given their values, norms, relationship with previous media technology shape or regulate the use of new media/communication tools so that their adaptation is acceptable to the group and fits the group's previous norms, values and religious and cultural practices. Campbell calls this expanded concept the religious social shaping of technology model (RSST). The American media researcher conducted a number of empirical studies in Christian, Muslim and Orthodox Jewish communities, where she mapped how religious norms set the boundaries of digital media use patterns. In the meantime, Campbell also developed the methodology of RSST research, based on which the researcher must clarify four issues during the investigation of a religious community. These are: (1) the history of the religious group you must be reviewed, (2) also the history of the religious group's previous media use; (3) what are their central values and religious convictions and finally (4) it is also necessary to explore how they frame their conversations and debates about the use of new technology in their everyday discourses. Based on these, the rules and norms according to which the religious communities will use the tools of the network media will be outlined. The theory of RSST is extremely useful in revealing how religious communities use the Internet and what kind of filters they implement. Campbell's RSST model provided the basis for many empirical studies (Andok 2021, 2023; Xu and Campbell 2021).

As we have seen, both theoretical frameworks emphasize and confirm with empirical research that religious media users will consider and filter when using the Internet. The criteria of the two frameworks partly overlap and partly differ. While Barzilai-Nahon and Barzilai examine the issues of hierarchy, patriarchy, disciple and seclusion, Campbell examines the history of the religious group, previous media use regulations, religious values and discourse. We must see that these aspects do not exclude or replace, but complement each other. And precisely because of this, they become suitable for grasping how a religious community shapes and filters the technology and contents of the Internet and how it is used in their everyday life. We must emphasize that the strongest aspect of the filter bubble was pointed out by the Brasilias, that religious users are only interested in information from their own community, and do not go online to find out about other religions and religious communities. From this point of view, in terms of religious information, we have to consider a strong bubble, a strong membrane. This is confirmed by the research of Zakaria and his colleagues who investigated the relationship between the filter bubble and religion in an Islamic religious environment in Indonesia (Zakaria et al. 2018). They also agree that in the perspective of religious phenomenology, technology is not neutral but human formation, as well as allowing the gap it can affect humans. …. At some point, the effect of this filter will produce a separate environment. The negative effect of the filter is the strengthening of a person's pretension to be reductive which leads to a radical attitude (Zakaria et al. 2018, p. 1). The significance of their analysis to show how the effect of a filter bubble impacts on the formation of religious attitudes. The filter bubble effect results in epistemological isolation and reduction for religious subjects leading to radicalism. Religious opinion no longer refers to the authority of the church, but the media. With regard to the filter bubble effect, both Zakaria and Barzilai-Nahon considered the process towards possible or real radicalization to be dangerous with regard to the Islamic religion (Barzilai-Nahon and Barzilai 2005, p. 14). "The Islamic religion in Indonesia, which is the majority religion of the people, along with the spread of media and radical ideology increases the attitude of fanaticism and radical. It is at this point that the significance of

media users' awareness of the dangers of media flows on the one hand and the effect of filter bubbles caused by our behaviour as users" (Zakaria et al. 2018, p. 7).

## 5. Guidelines of the Catholic Church Regarding Filter Bubble

A number of documents of the Catholic Church touched upon questions of media (Inter Mirifica 1963; Communio et Progressio 1971; Aetatis Novae 1991) and the use of the Internet (The Church and Internet 2002; Ethics in Internet 2002). The most recent was published on 28 May 2023, titeled *Towards Full Presence, Pastoral Reflection on Engagement with Social Media*, and it builds on the experiences of the Catholic Church during the coronavirus pandemic. When, due to physical closures, religious ceremonies could be held and broadcast online. One recent moment clearly demonstrated that digital media is a powerful tool for the Church's ministry. On 27 March 2020, while still in the early stages of the COVID-19 pandemic, Saint Peter's Square was physically empty but full of presence. The text is based on the words of The Parable of the Good Samaritan and shows who the neighbor is and who the stranger in the digital space is and how to behave with them according to the teachings of the Catholic Church. Thus, in 2023, the text begins with the statement that as individuals and as an ecclesial community, are to live in the digital world "... the question is no longer whether to engage with the digital world, but how. ... social media, which is one expression of digital culture, has had a profound impact on both our faith communities and our individual spiritual journeys" (Towards Full Presence 2023, sct. 2). But digital technology has made new kinds of human interactions possible and communication is increasingly influenced by artificial intelligence. The text encourages believers to become co-created spaces, not just something that we passively use. Examples of faithful and creative engagement on social media abound around the world, from both local communities as well as individuals who give witness to their faith on these platforms, often more pervasively than the institutional Church. There are also numerous pastoral and educational initiatives developed by local churches, movements, communities, congregations, universities and individuals. The Catholic Church takes a stand that network communication and social network sites can be linked to the public. Subsequently, the Church consolidated the image of social media as "spaces", not only "tools", and called for the Good News to be proclaimed also in digital environments. The digital world is a public sphere, a meeting place where believers can either encourage or demean one another, engage in a meaningful discussion or unfair attacks.

Like Sunstein and Pariser, the Vatican document also refers to the importance of experiences gained in the digital sphere and the possibility of showing religious values: "In the context of integrated communication, consisting of the convergence of communication processes, social media plays a decisive role as a forum in which our values, beliefs, language and assumptions about daily life are shaped. Moreover, for many people, especially those in developing countries, the only contact with digital communication is through social media. Well beyond the act of using social media as a tool, we are living in an ecosystem shaped at its core by the experience of social sharing. While we still use the web to search for information or entertainment, we turn to social network sites for a sense of belonging and affirmation, transforming it into a vital space where the communication of core values and beliefs takes place." Also, reflection on division is given an important place in the Catholic guidelines. "First of all, we are still dealing with a <digital divide>. Platforms that promise to build community and bring the world closer together have instead deepened various forms of division" (Towards Full Presence 2023, sct. 12).

After the division, the text deals with the phenomenon of the filter bubble itself. "Increasing emphasis on the distribution and trade of knowledge, data, and information has generated a paradox: in a society where information plays such an essential role, it is increasingly difficult to verify sources and the accuracy of the information that circulates digitally. ... The consequence of this increasingly sophisticated personalization of results is a forced exposure to partial information, which corroborates our own ideas, reinforces our beliefs, and thus leads us into an isolation of "filter bubbles". In a time when we

are increasingly divided, when each person retreats into his or her own filtered bubble, social media is becoming a path leading many towards indifference, polarization, and extremism" (Towards Full Presence 2023, sct. 19). The document also mentions that: people find common ground in gathering points against an external "other", a common ideological enemy. This kind of polarization yields a "digital tribalism" in which groups are pitted against others in an adversarial spirit. "For example, when groups that present themselves as "Catholic" use their social media presence to foster division, they are not behaving like a Christian community should. Instead of capitalizing on conflicts and adversarial click bait, hostile attitudes should become opportunities for conversion, an opportunity to witness encounter, dialogue, and reconciliation around seemingly divisive matters" (Towards Full Presence 2023, sct. 55). The guideline also sees and shows the negative consequences of the filter bubble phenomenon. "We have all witnessed automated systems that risk creating these individualistic "spaces", and at times encouraging extreme behaviors. Aggressive and negative speeches are easily and rapidly spread, offering a fertile field for violence, abuse, and misinformation" (Towards Full Presence 2023, sct. 16). Through the parable of the Good Samaritan, the document shows how, based on church values, one should behave in the digital sphere with the other, the stranger who belongs to another filter bubble. To welcome the "other", someone who takes positions opposed to my own or who seems "different", is certainly not an easy task. "The parable of the Good Samaritan, instead, challenges us to confront the digital "throw-away culture" and help each other to step out of our comfort zone by making a voluntary effort to reach out to the other. This is only possible if we empty ourselves, understanding that each one of us is part of wounded humanity, and remembering that someone has looked at us and had compassion on us" (Towards Full Presence 2023, sct. 21). The Vatican document details the dangers of network communication and social media, which rely on information taken out of context, while stating that the use of social media interfaces can be considered as "relationships with others and not just engagement in the exchange of information". And with this, the text actually takes a position in favor of the ritual model of communication (Carey 2009). It also talks about the problem of attention control, emotional reactions and the issue of authority: social media platforms are controlled by an external "authority", usually a for-profit organization that develops, manages and promotes changes to how the platform is programmed to work. In a broader sense, these all "live in" or contribute to the online "neighborhood." The question of the digital neighborhood comes up several times. By the way, Pariser already alluded to this: At its worst, the filter bubble confines us to our own information neighborhood, unable to see or explore the rest of the enormous world of possibilities that exist online (Pariser 2011, p. 68) The Vatican document also says: Social media "neighbors" are most clearly those with whom we maintain connections. At the same time, our neighbors are also often those we cannot see, either because platforms prevent us from seeing them or because they are simply not there. Digital environments are also shared by other participants such as "internet bots" and "deep fakes", automated computer programs that operate online with assigned tasks, often simulating human action or collecting data (Towards Full Presence 2023, sct. 21). According to the Catholic view: to be unneighborly on social media means being present to the stories of others, advocating for an integral vision of human. The text encourages believers to change the division, polarization, get out of the filter bubbles and make the digital public better by listening to others with understanding. "... We can change it. We can become drivers of change, imagining new models built on trust, transparency, equality, and inclusion. Together, we can urge media companies to reconsider their roles and let the internet become a truly public space. Well-structured public spaces are able to promote better social behavior. We need, therefore, to rebuild digital spaces so that they will become more human and healthier environments." It continues: ... We are invited to see the value and dignity of those with whom we have differences. We are also invited to look beyond our safety net, our silos, and our bubbles... And it all begins with the ability to listen well, to let the reality of the other touch us. The human person is made for relationship and community. At the

same time, loneliness and isolation plague our cultural reality, as we acutely experienced during the COVID-19 pandemic. ... We may be failing to provide space for those seeking to engage in. (Towards Full Presence 2023, sct. 54). So, we could see that the document published by the Catholic Church on Pentecost 2023, also shows the points where the current functioning of social media hides dangers in relation to the well-functioning public, it shows the connections between the filter bubble effect and polarization. And based on Catholic values, the task of the faithful is not to strengthen division, but to listen and help the other, the stranger, as the parable of the Good Samaritan teaches the faithful.

## 6. Conclusions

The aim of this study was to comprehensively and critically present the theoretical literature on online filtering in relation to religious contents their producers and users. The presentation of the literature horizon also created an opportunity to outline the theoretical framework in which the issue of religious filter bubbles and echo chambers can be discussed. Of course, this is not considering the two to be synonymous, since the filter bubble concept places a media technical element (personal settings) in the center, while the echo chamber focuses on connectivity and connectedness with like-minded people. To develop the framework, we took into account two types of filters related to media use, content creators and content recipients. In the former, gatekeeping and network gatekeeping, in the latter, selective exposure, Daily me, the echo chamber and the filter bubble phenomenon and their criticism were presented, also referring to religious communities. Two of the theories of recipient selection are highlighted in the study. One is Barzilai-Nahon and Barzilai's cultured technology theory, the other is Heidi Campbell's Religious Social Shaping of Technology theory. Both point to the aspects that need to be taken into account in media technology and media content filtering by religious users. These are the following: hierarchy, patriarchy, disciple and seclusion, the history of the religious group's previous media usage regulations, its religious values and the way the discourse is conducted about it. Following their suggestions, the study addressed the guidelines of the Catholic Church and highlighted the aspects of the Towards Full Presence (2023) document, which talk about the dangers of the filter bubble and encourage Christians to dismantle it.

**Funding:** This research received no external funding.

**Institutional Review Board Statement:** The study did not require ethical approval.

**Informed Consent Statement:** Not applicable.

**Data Availability Statement:** No new data were created.

**Conflicts of Interest:** The author declare no conflict of interest.

## Note

1      Danah boyd herself lowercases her name. see https://www.danah.org/ (accessed on 19 October 2023).

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
