# Peer review of "Religious Filter Bubbles on Digital Public Sphere"

_religions, doi:10.3390/rel14111359_

Round 1

Reviewer 1 Report

Comments and Suggestions for Authors

The authors of this essay do a great job of detailing the media literature regarding gatekeepers of information, particularly the news, and how digital media allows for content filtering and creation at the individual level as well as the organizational and institutional level. Before focusing on filter bubbles, the authors go into selective exposure of media and how it creates echo chambers. From there the authors discuss religious filter bubbles and identify the manifestation of filter bubbles in a religious context (e.g., Catholic Church). The real value of the essay comes on page 14 where the author(s) leverage the parable of the Good Samaritan to unpack the manifestation of religious filter bubbles and explains society’s need to work with others who do not share our same views. Brilliant!

The arguments about media filtering (gatekeeping and echo chambers) are clear and logical, but the filter bubble argument is lacking (see first half of page 7). Perhaps this is because of the heavy use of Pariser (2011) quotes, which dominate the first half of page 7. These quotes don’t necessarily lend themselves to the author(s) conversational style of writing. Obviously, the author(s) unpack Pariser in the latter half of page 7, but the initial arguments could be laid out a bit better.     

The paper is well written and includes logical arguments. The author(s) do a good job of explaining the views of their sources; however, on occasion the author(s) could strengthen their arguments by providing more sources to back up their insights. For example, when discussing trust in news created by amateurs, it would be worth contextualizing the decrease in overall trust in organizations, per the Edelman Trust Barometer, instead of solely relying on Pariser (2011) on p. 6. Essentially, identifying a few other relevant sources to the trust arguments would lend credibility to the arguments and strengthen the paper.

When doing a final edit, double check the spelling of some of your sources to ensure the references match the intext citations. Also, Shoemaker should be referred to as “she”, not “he” (see Shoemaker section on page 3). On p. 15 line 742 you stated, “the latest document of the Catholic Church;” however, this will not be the “latest document” 10 years from now. Adjust some of the language within the essay as your essay has the potential to become an evergreen piece of scholarship referred to by “media and religion” scholars as well as media AND religion scholars years from now.

The abstract focuses too much on the first half of the paper, which lays out the media concepts. Focus the abstract on the true value of this essay, which is the latter half of the paper: the value found in religious filter bubbles as well as the challenges with them. Mentioning the Catholic Church example would also enhance the abstract. Essentially, make sure the abstract reflects the true value of the essay, and especially the value of this essay to this journal. It’s a bit lack luster. The current abstract could really be part of the introduction or set the stage for part 2.

Author Response

Dear Reviewer,

Thank you very much for your comments. It was all useful and I took the advice.

First of all, I transformed the abstract so that the content of the second half of the study also appears in it as a summary.

Then I changed the part based on Eli Pariser's book, since the style of the quotes is really far from the style of the study. I incorporated the data of the Edelman Trust Barometer suggested by you and also supplemented it with the data of the Digital News Project. I refer to other critical texts regarding the filter bubble theory.

I corrected the references to Pamela Shoemaker from he to she. Thank you very much for noticing this mistake, it is very unpleasant that I referred to her as a man in the text.

Finally, I do not refer to the Vatican document as the last document, but according to the date of its publication, so that it will not be misleading later.

Thank you very much for taking the time to read and review this study. To be honest, I rarely receive such insightful, benevolent and forward-thinking criticism.

Sincerely,

Author

Reviewer 2 Report

Comments and Suggestions for Authors

- Are the concepts discussed applicable to all religious communities, both monotheistic and non-monotheistic?

- An exhaustive study implies that all communities and their functioning in relation to filters are examined.

- How are religious contents studied?

Comments on the Quality of English Language

- English language fine and clear

Author Response

Dear Reviewer,

Thank you very much for your comments and I can answer the following questions:

Are the concepts discussed applicable to all religious communities, both monotheistic and non-monotheistic?

Yes, it can be used, because the filter bubble effect can be linked to network communication and not to the difference of religions. Rather, the question is whether or not the religious community makes use of networked communication and social media. If so, the filter bubble effect will more or less apply to it.

- An exhaustive study implies that all communities and their functioning in relation to filters are examined.

No, of course I couldn't examine every community in social media, but I could examine different types.

- How are religious contents studied?

We examine religious content using the content analysis method. But this current research did not focus on the contents themselves, but on filtering.

Thank you very much for your comments.

Sincerely,

Author